# Diagnostic value of the molecular detection of *Sarcoptes scabiei* from a skin scraping in patients with suspected scabies

**Moonsuk Bae**[1☯], **Ji Yeun Kim**[1☯], **Jiwon Jung**[1], **Hye Hee Cha**[1], **Na-Young Jeon**[1], **Hyun-Jung Lee**[1], **Min Jae Kim**[1], **Sung Eun Chang**[2]*, **Sung-Han Kim**[1]*

**1** Department of Infectious Diseases, Asan Medical Center, University of Ulsan College of Medicine, Seoul, Republic of Korea, **2** Department of Dermatology, Asan Medical Center, University of Ulsan College of Medicine, Seoul, Republic of Korea

☯ These authors contributed equally to this work.
* csesnumd@gmail.com (SEC); kimsunghanmd@hotmail.com (S-HK)

**Data Availability Statement:** All relevant data are within the manuscript and its Supporting Information files.

## Abstract

Scabies is a highly contagious parasitic disease associated with long-term residence in nursing homes, and it is a public health burden worldwide. However, atypical skin manifestations are frequent and the widely used diagnostic test based on microscopic examinations has limited sensitivity. We evaluated the diagnostic value of polymerase chain reaction (PCR) from skin scraping in patients with suspected scabies. Adult patients with suspected scabies, unrelated diseases or healthy volunteers were enrolled at a tertiary hospital, in Seoul, South Korea, from December 2017 through October 2018. We classified participants based on the consensus criteria established by the International Alliance for the Control of Scabies in 2018; confirmed (microscopic mite detection), clinical (scabies burrow or typical lesions with two history features including itch and close contact with scabies patients), suspected scabies (typical lesion with one history feature or atypical lesion with two history features), or no scabies. PCR was performed on the skin scrapings to target the *cytochrome c oxidase subunit 1* (*cox1*) gene of *Sarcoptes scabiei*. A total of 47 participants, 33 with suspected scabies, 10 with unrelated diseases, and 4 healthy volunteers were enrolled. Of the 33 patients, 22 were classified as confirmed scabies, 2 as clinical scabies, 6 as suspected scabies, and 3 as no scabies. The sensitivities of the microscopic examination were 100%, 92%, and 73% in confirmed scabies; confirmed and clinical scabies; and confirmed, clinical, and suspected scabies, respectively ($p = 0.006$). The sensitivities of PCR were 86%, 83%, and 80% in confirmed scabies; confirmed and clinical scabies; and confirmed, clinical, and suspected scabies, respectively ($p = 0.59$). The specificity of the scabies PCR in the no scabies control was 100% (95% CI = 80–100). PCR testing for scabies may be helpful in the improvement of sensitivity for the diagnosis of scabies by clinical criteria.

**Funding:** This work was supported by grants from Korean Society for Chemotherapy (http://www.ksat.or.kr/) to JJ in 2018 and the Korea Health Technology R&D Project through the Korea Health Industry Development Institute, funded by the Ministry of Health & Welfare, Republic of Korea to S-HK (grant no. HI16C0272). The funders had no role in study design, data collection and analysis, decision to publish, or preparation of the manuscript.

**Competing interests:** All authors report no conflicts of interest relevant to this article.

## Author summary

Scabies occasionally presents in atypical forms causing a delay in diagnosis, which can lead to the outbreaks in residential and nursing care for elderly people. We hypothesized that polymerase chain reaction (PCR) detection of *Sarcoptes scabiei* DNA directly has higher sensitivity than microscopic examination. Recently, clinical consensus criteria have been proposed by the International Alliance for the Control of Scabies (IACS) to overcome the low sensitivities of conventional diagnostic tests for scabies. We thus evaluated the diagnostic capability of in-house real-time PCR for the diagnosis of scabies from skin scraping in subjects with suspected scabies and with unrelated disease according to the criteria of the IACS. We found that the diagnostic sensitivity of scabies PCR maintained between 86% and 80% as the level of diagnostic certainty by the IACS criteria decreased, while the diagnostic sensitivity of microscopic examinations decreased from 100% to 73% as the level of diagnostic certainty by the IACS criteria decreased. Our data suggested that our in-house scabies PCR test was a useful adjunct in the improvement for the diagnosis of scabies by the consensus criteria.

## Introduction

Scabies caused by *Sarcoptes scabiei* mites, is a highly contagious parasitic disease characterized by intense itching which is aggravated at night. Infections by scabies mite result in various skin eruptions consisting of papules, nodules, vesicles, and excoriated eczematous lesions due to scratching. The skin lesions involve intertriginous areas including the finger webs, the wrists, axillae, buttocks, genitals, and the breasts (in females only) [1]. The clinical features of scabies in the elderly may differ from those in younger individuals [2]. Transmission of scabies occurs predominantly through close and prolonged contact. Nosocomial outbreaks are usually associated with long-term residence in nursing homes and represent a tremendously increasing problem in high-income countries [3]. One of the most important risk factors for nosocomial outbreaks is the failure to recognize scabies in patients by attending clinicians. The crusted scabies has particularly atypical presentation making it difficult to diagnose and high risk of transmission, therefore it has been the index case in most recent outbreaks [4]. The misdiagnosis of scabies is common due to its multifarious manifestation and the absence of available and appropriate diagnostic laboratory tests [5]. Recently, clinical consensus criteria have been proposed by the International Alliance for the Control of Scabies (IACS) to overcome the low sensitivities of conventional diagnostic tests for scabies [6]. However, few studies have evaluated the diagnostic performance of these tests.

Microscopic examination of skin scrapings is widely used for the diagnosis of scabies, but it has a suboptimal sensitivity of only 50% [7]. Dermoscopy has been widely used, but it has a disadvantage in that it has low specificity and is affected by the dermatologist's experience [7]. More sensitive and specific diagnostic tests are urgently required. Polymerase chain reaction (PCR) has been studied for the diagnosis of scabies, offering higher sensitivity than conventional microscopy [8–12]. We developed a novel real-time PCR assay for *cytochrome c oxidase subunit 1* (*cox1*) gene, which is relatively conserved and has no known homology with other common human ectoparasites. We evaluated the diagnostic capability of this in-house real-time PCR for the diagnosis of scabies from skin scraping in subjects with suspected scabies and with unrelated disease according to the previously used criteria of the IACS.

## Method

### Patients and clinical diagnosis of scabies

Adult patients with suspected scabies or other unrelated diseases, such as onychomycosis, or control healthy volunteers were enrolled between December 2017 and October 2018. This prospective cross-sectional study was performed at a 2700-bed tertiary-care teaching hospital in Seoul, Republic of Korea. As a first step, an experienced dermatologist delineated clinically suspicious lesions. Skin scrapings for patients with suspected scabies were obtained from clinically suspected lesions by an experienced nurse specialist for microscopic examination and scabies PCR simultaneously [13]. The specimens were collected at 10~15 sites for all delineated lesions. Skin scrapings for healthy volunteers were obtained from extensor sites of extremities, which were appropriate places in getting keratinous layer without skin lesions. Decision regarding the appropriate anti-scabies treatment was made based on the clinical features and microscopic results and the blinded results of the scabies PCR by the attending dermatologist. Only the results of scabies PCR and microscopic examination for the specimens obtained on the first day of diagnosis with scabies were included in the analysis.

We classified participants based on the consensus criteria established by the IACS in 2018 [6]: confirmed (mites, eggs, or feces on microscopic examination of skin scrapings); clinical (scabies burrow, typical lesions affecting the male genitalia, or lesions in a typical distribution with two history features including itch and close contact with an individual who had clinical scabies); suspected scabies (typical lesion with one history feature or atypical lesion with two history features); or no scabies. Imaging devices such as dermoscopes were not used. Typical lesions were defined as multiple small papules, nodules, vesicles, or excoriation, and a typical distribution of lesions was defined on observation of the lesions on the finger-webs, wrists, hands, axillae, gluteus, genitals, or breasts in the case of female patients [1].

### DNA extraction and real-time PCR

Samples obtained via skin scraping were collected into and stored at −80˚C until PCR analysis was performed. DNA extraction was performed using the QIAamp DNA mini kit (Qiagen, Hilden, Germany) according to manufacturer's instructions, with some modifications. In brief, skin samples were placed in a sterile microcentrifuge tube and digested with 40 μL of 1 M dithiothreitol (DTT, Sigma-Aldrich, Inc, St. Louis, Missouri, USA), 300 μL of ATL buffer, and 40 μL of proteinase K (Qiagen) at 56˚C for 2 h. Lysates were incubated with 380 μL of AL buffer (Qiagen) at 70˚C for 10 min. After incubation, 380 μL of 100% ethanol (Sigma-Aldrich) were added and further extraction and purification procedures were followed according to the manufacturer's protocols. Purified DNA was eluted in 100 μL of AE buffer (Qiagen) and used for PCR amplification.

To detect scabies, primers and probe were designed for the *cox1* gene of *Sarcoptes scabiei* (GenBank accession number: KX827306.1). A 196 fragment of the highly conserved region was amplified using the specific primers cox1F (5'- ATGATTTCTATTGCAACTTTAGG-3') and cox1R (5'- TTGCTCAATACATAGAGGGGTTA-3'). Taqman probe cox1P2 (5'-FAM-AATATTAGGGGGAAAATTAGATTTTAACCC-BHQ1-3') was used for real-time detection. The specificity of each primer and probe was checked using BLAST search against the NCBI database, and no homologies between the designed primers and probe for *S. scabiei* and other mite species such as *Demodex* sp., *Dermatophagoides* sp., *Cheyletus* sp., and *Thyrophagus puterescentiae* were found. The internal control was amplified and detected using ACBT_F (5'-ACTAACACTGGCTCGTGTGA-3'), ACBT_R (5'- CTTGGGATGGGGAGTCTGTT-3'), and ACBT_P (5'-HEX- AGGCTGGTGTAAAGCGGCCTTGG-BHQ1-3').

Multiplex real-time PCR assays to detect *S. scabiei* were performed using the LightCycler FastStart Essential DNA Probe Master (Roche, Basel, Switzerland) in 20 μL of reaction mixtures containing 10 μL of 2X master mix, 3.1 μL of primer and probe mix consisting of 500 nM cox1 primers, 150 nM cox1 probe, 150 nM ACBT primers and 100 nM ACBT probe, and 5 μL of extracted DNA. Real-time PCR amplification was performed with the LightCycler 96 system (Roche) in the following conditions: pre-incubation at 95˚C for 5 m, followed by 45 cycles of a 2-step amplification (95˚C for 5 s and 56˚C for 30 s). Each assay included a positive and a negative control. The PCR product was electrophoresed and purified using the QIAquick gel extraction kit (Qiagen) (Fig 1). The purified PCR product was sequenced and the DNA sequences confirmed their identity through a BLASTn search in NCBI.

## Generation of calibration curves and confirmation of specificity

PCR products identified as the *cox1* gene of *S. scabiei* were inserted into a plasmid vector (T-Blunt PCR cloning kit, Solgent, Daejeon, Korea) and used as standards for quantification and positive controls. The standard plasmids were quantified with a Nanodrop spectrophotometer (Thermo Scientific) and 7 serial 10-fold dilutions of the plasmid, ranging from $10^6$ to $10^0$ copies, were amplified using multiple real-time PCR. The correlation curves were determined by plotting the Ct values against the log of the copy number (Fig 2). The analytical sensitivity of the real-time PCR assay was one copy per reaction.

We further evaluated the potential risks of cross-reaction with this PCR. As a negative control, DNA extracted from ground bodies of house dust mites (*Dermatophagoides pteronyssinus* and *Dermatophagoides farinae*) was also used to test the analytic specificity of the novel PCR. All DNA samples from specimens with *Dermatophagoides pteronyssinus* and *Dermatophagoides farinae* were negative for the PCR using the *cox1* gene primer and probe set.

## Statistical analysis

To investigate the diagnostic value of the PCR testing, we calculated the 95% confidence interval (CI) of the sensitivity and specificity of both PCR and microscopy assays for each clinical category according to the level of diagnostic certainty: confirmed; confirmed and clinical; and confirmed, clinical, and suspected scabies. The sensitivity of the scabies PCR was compared with those of microscopic examination in each clinical category group. Data are expressed as percent and Clopper-Pearson exact 95% CI. The linear-by-linear association test was used to examine the linear association between the level of diagnostic certainty and the sensitivity of the microscopic examination or the scabies PCR test. The software used for statistical analyses was SPSS v. 24.0 (IBM, Armonk, New York, USA).

## Ethics statement

Written informed consent was obtained from all patients. The study's approval was obtained from the Institutional Review Board of the Asan Medical Center (2018–0640).

## Results

A total of 47 participants including 33 patients with suspected scabies and 14 participants with an unrelated disease (10 patients with onychomycosis and 4 healthy volunteers) were enrolled in this study. Of the 33 patients with suspected scabies, 22 were classified as confirmed scabies, 2 as clinical scabies, 6 as suspected scabies, and 3 as no scabies (2 as drug eruption, 1 as another superficial fungal infection). A total of 17 subjects including 3 with no scabies, 10 with onychomycosis, and 4 healthy volunteers were classified as no scabies control.

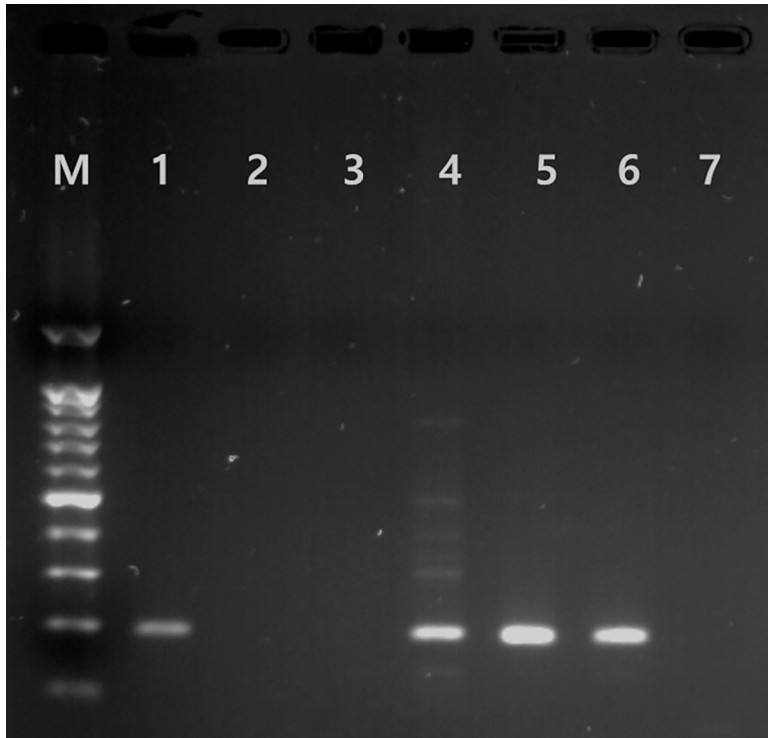

**Fig 1. Detection of *Sarcoptes scabiei* DNA by polymerase chain reaction (PCR).** DNA was purified from skin scraping samples from suspected scabies patients (lanes 1–5), and run by electrophoresis with a positive control (lane 6), and a negative control (lane 7). Lane M is a 100-bp DNA ladder marker.

The median age of the patients at the time of diagnosis in the 30 scabies patients was 60±17 years (95% CI = 54–67) and 57% were male. Thirteen of these patients (43%) had a history of hospitalizations and 10 (33%) had a history of hospital visits associated with family or friends. Seven patients (23%) were bedridden resulting from other debilitating medical conditions and 1 patient (3%) did not bring the itching symptoms to the physician's attention due to other previously diagnosed neurological diseases. Overall, the median time from symptoms onset to diagnosis of scabies was 77 days (95% CI = 47–107). The most common skin manifestation was papules in 23 patients (77%), followed by nodules in 2 (7%) and excoriated patches in 2 (7%). There was no case reported of crusted scabies. The torso was the most common location in 27 patients (90%), followed by the extremities in 22 (73%), and the genitalia in 7 (23%) of the patients. Approximately 43% of patients with scabies had the skin lesion in the hands (S1 Fig).

The diagnostic performances of PCR and microscopy are shown in Table 1. The sensitivity of the microscopic examination was 100% (95% CI = 85–100; 22/22), 92% (95% CI = 73–99; 22/24), and 73% (95% CI = 54–88; 22/30) in confirmed scabies; confirmed and clinical scabies; and confirmed, clinical, and suspected scabies, respectively ($p = 0.006$). As the level of diagnostic certainty decreased, the sensitivity of microscopy decreased. The microscopic examination specificity was 100% (95% CI = 80–100; 17/17). The sensitivity of the scabies PCR test was 86% (95% CI = 65–97; 19/22), 83% (95% CI = 63–95; 20/24), and 80% (95% CI = 61–92; 24/30) in confirmed scabies; confirmed and clinical scabies; and confirmed, clinical, and suspected scabies, respectively ($p = 0.59$). As the level of diagnostic certainty decreased, the sensitivity of the scabies PCR also decreased, but there was no statistically significant difference between the

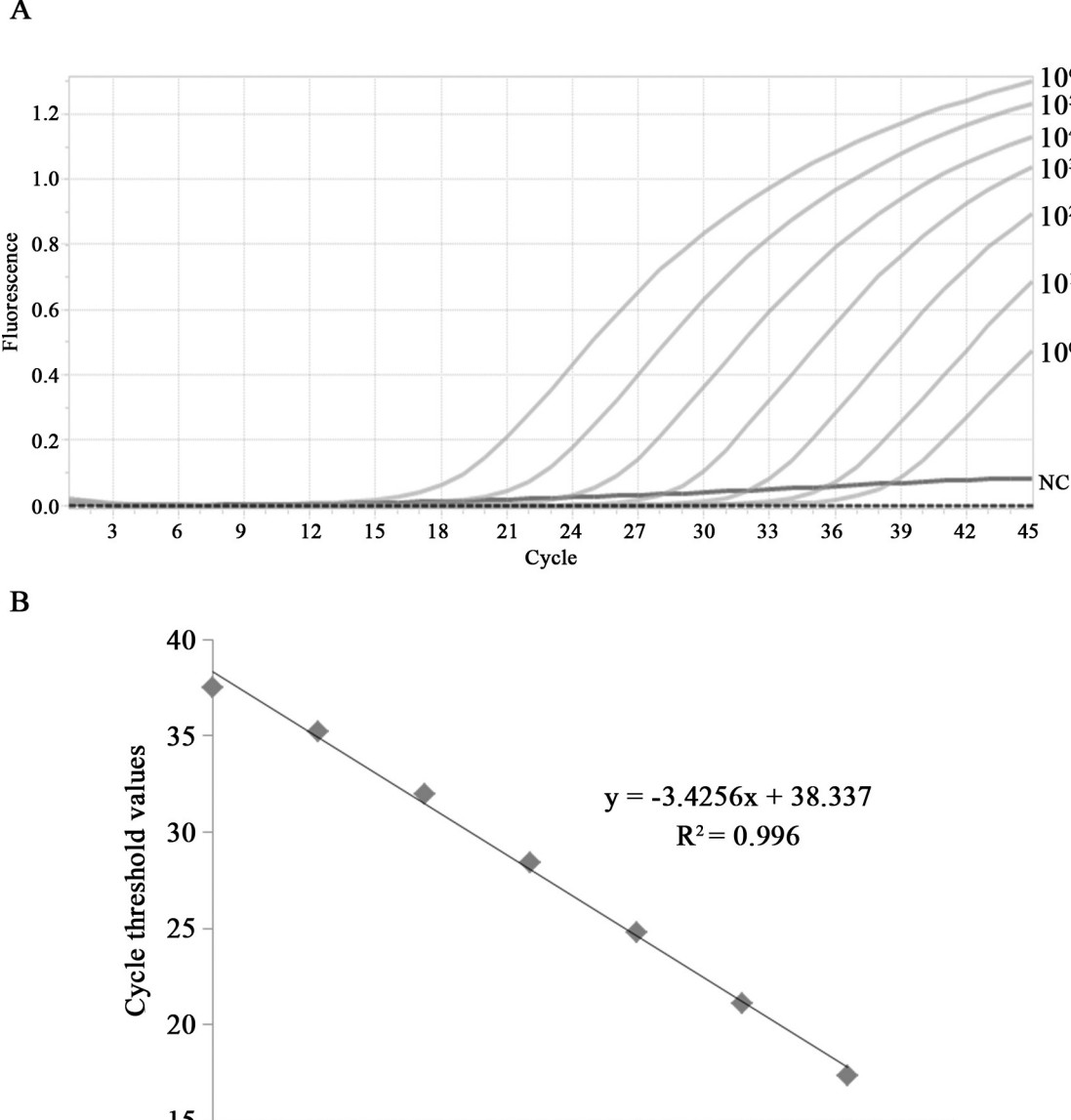

**Fig 2. Correlation curves for real-time PCR detection of scabies.** (A) Amplification curves and (B) correlation curves of standard DNA ranging from $10^6$ to $10^0$ copies.

groups. The sensitivity of scabies PCR test was slightly higher than that of the microscopic examination but there was no statistically significant difference between these tests in patients with confirmed, clinical, and suspected scabies (80 vs 73%, $p = 0.54$). Of the 30 patients with scabies, 5 (17%) revealed negative results for microscopic examination, but positive results for scabies PCR. They were composed of 1 patient with clinical scabies and 4 patients with suspected scabies. On the other hand, 3 patients (10%) exhibited positive results for microscopic examination, but negative results for scabies PCR. Skin scraping specimens from 10 patients with onychomycosis, 4 healthy donors without any other dermatologic condition, and 3 classified as no scabies by the IACS criteria were all negative for the PCR test. Thus, the specificity

**Table 1. Comparison of diagnostic performance of the scabies PCR and microscopy in confirmed, clinical, and suspected scabies patients.**

| | | Confirmed scabies (n = 22) | Confirmed + clinical scabies (n = 24) | Confirmed + clinical + suspected scabies (n = 30) | No scabies (n = 17)[a] |
|---|---|---|---|---|---|
| **PCR** | Positive (n) | 19 | 20 | 24 | 0 |
| | Negative (n) | 3 | 4 | 6 | 17 |
| | Sensitivity (%, 95% CI)[b] | 86 (65–97)[d] | 83 (63–95)[e] | 80 (61–92)[f] | |
| | Specificity (%, 95% CI) | | | | 100 (80–100) |
| **Microscopy** | Positive (n) | 22 | 22 | 22 | 0 |
| | Negative (n) | 0 | 2 | 8 | 17 |
| | Sensitivity (%, 95% CI)[c] | 100 (85–100)[d] | 92 (73–99)[e] | 73 (54–88)[f] | |
| | Specificity (%, 95% CI) | | | | 100 (80–100) |

[a]"No scabies" included 13 skin scrap specimens from patients with alternative diagnosis and four skin scrap specimens from healthy volunteers.

[b]Difference in the sensitivity of the PCR test between the three groups was not statistically significant ($p = 0.59$).

[c]Difference in the sensitivity of the microscopic examination between the three groups was statistically significant ($p = 0.006$).

[d]Difference in the sensitivity between PCR and the microscopic examination in the patients with confirmed scabies was not statistically significant ($p = 0.23$).

[e]Difference in the sensitivity between PCR and the microscopic examination in the patients with confirmed and clinical scabies was not statistically significant ($p = 0.67$).

[f]Difference in the sensitivity between PCR and the microscopic examination in the patients with confirmed, clinical, and suspected scabies was not statistically significant ($p = 0.54$).

of the scabies PCR in no scabies controls was of 100% (95% CI = 80–100; 17/17). There was no statistically significant difference in the PCR sensitivity according to gender, age, location of skin lesion, and type of skin lesion.

## Discussion

According to previous reports, the sensitivity of the scabies PCR ranged between 30% and 60% based on clinically suspected scabies [8–12]. However, there was no appropriate reference standard test, thus microscopic examination used as a reference standard in these studies, which limits the diagnostic usefulness of the scabies PCR test. To overcome this limitation, we applied the consensus criteria for the diagnosis of scabies developed by the IACS [6] in our study. We categorized each group according to the level of diagnostic certainty and compared the sensitivity of the microscopic examination and the scabies PCR test for each group. Consequently, we found that the sensitivity of the microscopic examination and the scabies PCR test tended to decrease as the level of diagnostic certainty decreased, although the decreasing sensitivity of the microscopic examination was more prominent than that of the scabies PCR. Possible explanations for this observation are that patients with clinical or suspected scabies by the IACS criteria did not have scabies or revealed false-negative results for the microscopic examination due to its low sensitivity. However, given the high specificity of the scabies PCR in our study and previous consistent reports on the low sensitivity of the microscopic examination, the scabies PCR was considered to have an ability to identify cases that could not be confirmed by microscopic examination. Our findings support this hypothesis and showed that the scabies PCR test detected five additional patients (17%) that were not initially diagnosed by microscopic examination. However, in three patients, the microscopic examination results were positive and the scabies PCR results were negative. This discrepancy could have been due to the genetic diversity of *S. scabiei*. In a recent study, it was suggested that *S. scabiei* mites in humans were mainly distributed into three genetically distinct clades [14]. The authors performed sequencing of *cox1* gene obtained in mites from 60 humans and showed that mites could

belong to different clades genetically even if they were collected in the same area. Taken together, despite no significant difference, the detection rate of scabies by PCR tended to be higher than that of the microscopy examination, suggesting the high sensitivity of scabies PCR.

In this study, the topographic distribution of skin lesions in patients with scabies presented similarly in a recent study [2]. The torso was the most common location involving scabies. The proportion of hand (43%) and genitalia (23%) was relatively higher than that in the recent study (36% and 7%) [2]. The possible explanation for this observation might be that this study mainly included symptomatic patients in the non-outbreak situation, unlike the recent study based on the nosocomial outbreaks [2]. It is worth to note that there are limited data on whether the sensitivity of the microscopic examination depends on the location of skin lesions. We found that there was no statistically significant difference in the scabies PCR sensitivity according to the locations of skin lesions. However, further studies are needed on this area.

There are some potential limitations to our study. First, the consensus criteria for the diagnosis of scabies [6] was still reliant upon expert opinion. Further studies will be required to understand if these criteria can be used to evaluate the clinical potential of new diagnostic tests. Second, skin scraping for scabies PCR was required and depended on the expertise of an experienced specialist for obtaining good-quality skin samples. Further studies are needed on the diagnostic performance of various tests for scabies depending on the expertise of skin scraping. Third, this study included a small number of patients, which potentially accounts for the lack of difference in the sensitivity between the microscopy examination and the scabies PCR test.

In conclusion, scabies PCR was shown to offer an improvement in assay sensitivity compared to that of microscopy examination for the diagnosis of scabies by clinical criteria. This technique can, therefore, be considered as an adjunct method for the diagnosis of scabies, particularly in microscopy-negative suspected cases. Further larger-scale studies will be needed to evaluate the diagnostic performance of the scabies PCR and to validate the new IACS criteria by using more sensitive diagnostic tests.

## Supporting information

**S1 Fig. Distribution of skin lesions.** Percentage in the specific locations referred to the proportion of patients with scabies who presented skin lesion at that location.
(TIF)

## Author Contributions

**Conceptualization:** Sung-Han Kim.

**Data curation:** Moonsuk Bae, Ji Yeun Kim, Jiwon Jung, Hye Hee Cha, Na-Young Jeon, Hyun-Jung Lee.

**Formal analysis:** Moonsuk Bae, Ji Yeun Kim, Jiwon Jung.

**Funding acquisition:** Jiwon Jung.

**Investigation:** Ji Yeun Kim, Hye Hee Cha.

**Methodology:** Moonsuk Bae, Ji Yeun Kim, Hye Hee Cha.

**Supervision:** Sung Eun Chang, Sung-Han Kim.

**Validation:** Moonsuk Bae, Ji Yeun Kim.

**Writing – original draft:** Moonsuk Bae, Ji Yeun Kim, Sung-Han Kim.

**Writing – review & editing:** Jiwon Jung, Min Jae Kim, Sung Eun Chang.

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
