## [Decision Letter · Decision Letter 0]

11 Jan 2020

Dear Dr. Kim,

Thank you very much for submitting your manuscript "Diagnostic value of the molecular detection of Sarcoptes scabiei from a skin scraping in patients with suspected scabies" (#PNTD-D-19-01821) for review by PLOS Neglected Tropical Diseases. Your manuscript was fully evaluated at the editorial level and by independent peer reviewers. The reviewers appreciated the attention to an important problem, but raised some substantial concerns about the manuscript as it currently stands. These issues must be addressed before we would be willing to consider a revised version of your study. We cannot, of course, promise publication at that time.

We therefore ask you to modify the manuscript according to the review recommendations before we can consider your manuscript for acceptance. Your revisions should address the specific points made by each reviewer. 

When you are ready to resubmit, please be prepared to upload the following:

(1) A letter containing a detailed list of your responses to the review comments and a description of the changes you have made in the manuscript.

(2) Two versions of the manuscript: one with either highlights or tracked changes denoting where the text has been changed (uploaded as a "Revised Article with Changes Highlighted" file); the other a clean version (uploaded as the article file).

(3) If available, a striking still image (a new image if one is available or an existing one from within your manuscript). If your manuscript is accepted for publication, this image may be featured on our website. Images should ideally be high resolution, eye-catching, single panel images; where one is available, please use 'add file' at the time of resubmission and select 'striking image' as the file type. 

Please provide a short caption, including credits, uploaded as a separate "Other" file. If your image is from someone other than yourself, please ensure that the artist has read and agreed to the terms and conditions of the Creative Commons Attribution License at http://journals.plos.org/plosntds/s/content-license (NOTE: we cannot publish copyrighted images). 

(4) If applicable, we encourage you to add a list of accession numbers/ID numbers for genes and proteins mentioned in the text (these should be listed as a paragraph at the end of the manuscript). You can supply accession numbers for any database, so long as the database is publicly accessible and stable. Examples include LocusLink and SwissProt.

(5) To enhance the reproducibility of your results, we recommend that you deposit your laboratory protocols in protocols.io, where a protocol can be assigned its own identifier (DOI) such that it can be cited independently in the future. For instructions see http://journals.plos.org/plosntds/s/submission-guidelines#loc-methods

While revising your submission, please upload your figure files to the Preflight Analysis and Conversion Engine (PACE) digital diagnostic tool, https://pacev2.apexcovantage.com/ PACE helps ensure that figures meet PLOS requirements. To use PACE, you must first register as a user. Then, login and navigate to the UPLOAD tab, where you will find detailed instructions on how to use the tool. If you encounter any issues or have any questions when using PACE, please email us at figures@plos.org.

We hope to receive your revised manuscript by Mar 11 2020 11:59PM. If you anticipate any delay in its return, we ask that you let us know the expected resubmission date by replying to this email.

To submit a revision, go to https://www.editorialmanager.com/pntd/ and log in as an Author. You will see a menu item call Submission Needing Revision. You will find your submission record there. 

Sincerely,

Kosta Y. Mumcuoglu, PhD

Associate Editor

Paul J. Brindley, PhD

Editor-in-Chief

Line 36: Do not use abbreviations such as "exam"

Line 38: Place a full stop at the end of the sentence

Line 62: scabiei rather than scabies

Line 63-65: The sum of the patients mentioned is not 43 (see also lines 195-197)

Line 83: Mention also the crusted scabies

LIne 90: Mention also the dermoscope

Line 232: Mention the number as 13.

Reviewer's Responses to Questions

**Key Review Criteria Required for Acceptance?**

**Methods**

-Are the objectives of the study clearly articulated with a clear testable hypothesis stated?

-Is the study design appropriate to address the stated objectives?

-Is the population clearly described and appropriate for the hypothesis being tested?

-Is the sample size sufficient to ensure adequate power to address the hypothesis being tested?

-Were correct statistical analysis used to support conclusions?

-Are there concerns about ethical or regulatory requirements being met?

Reviewer #1: Please add no and date of ethical approval

Reviewer #2: The study design was appropriate to address the stated objectives, with the novel use of IACS consensus criteria as ‘gold standard’, allowing sensitivity for PCR and microscopy to be calculated independently of each other. Additional information, and correction of some details is required, as follows:

1) Please double check the numbers in each of the categories throughout the study, as follows:

Line 62-65: The numbers of participants in each group is inconsistent with the total number of participants, as well as differing from the description given later in the paper. I.e. 43 suspected +14 other diseases+4 healthy = 61, not 57; 22 confirmed + 2 clinical + 6 suspected + 3 no scabies=33 suspected scabies cases, not 43. 

Line 193-194: Here, the 14 participants without suspected scabies is broken down into 10 with onychomychosis and 4 healthy, inconsistent with the abstract in which the 4 healthy were stated as additional to the 14.

Line 195-196: As per the abstract, the numbers in each category of scabies do not add to the total stated with suspected scabies.

Line 198-200: With 17 controls (10 with onychomychosis, 3 with no scabies, 4 healthy volunteers) and 30 scabies diagnoses, it appears as though the total number of participants was 47 rather than 57.

2) Where were the skin scrapings taken from for the healthy volunteers with no scabies? To ensure comparability with the scrapings taken from suspected cases, the sampling sites should be similar. Please state the sampling strategy for the controls in the methods. 

3) Line 182-183: Could you please state that you have calculated ‘exact’ confidence intervals (your results in Table 1 appear to be consistent with this).

**Results**

-Does the analysis presented match the analysis plan?

-Are the results clearly and completely presented?

-Are the figures (Tables, Images) of sufficient quality for clarity?

Reviewer #1: -Are there any differences for the results according to gender or age?

-Are there any differences for the results according to the distribution of lesions (finger-webs, wrists, hands, axillae, gluteus, genitals, or breasts or may be mixed?)?

-Are there any differences for the results according to lesions (multiple small papules, nodules, vesicles, or

excoriation or others ???)? 

Please add your results (Could be the reason for getting different results).

-Samples collected for microscopy and PCR were from the same place? Give details (Could be the reason for getting different results).

Do you search only one sample or you repeated?? 

New statistical analyses could be required for lesion distribution and type..

For 222-225

"Of the 30 patients with scabies, 5 (17%) revealed negative results for microscopic examination, but positive results for scabies PCR". Which type of scabies? Confirmed, clinical, or suspected?

"On the other contrary, 3 patients (10%) exhibited positive results for microscopic examination, but negative results for scabies PCR". Which type of scabies? Confirmed, clinical, or suspected?

Reviewer #2: The results presented follow the outlined methods, with tables and figures clearly presented. There are minor corrections required to the numbers, with details provided under the methods section.

**Conclusions**

-Are the conclusions supported by the data presented?

-Are the limitations of analysis clearly described?

-Do the authors discuss how these data can be helpful to advance our understanding of the topic under study?

-Is public health relevance addressed?

Reviewer #1: Additional evaluation mentioned in the result section that should be discussed in discussion section

Reviewer #2: The conclusions are supported by the experimental results and the limitations of the study are clearly described. Further work required is described. A couple of sentences should be clarified, as follows:

1) Line 254-255: ‘clinical or suspected cases is more likely to be detected by the scabies PCR’ – this needs to be reworded. By design, clinical or suspected cases cannot be positive for microscopy – because if they were, then they would be confirmed cases. Therefore, it doesn’t make sense to compare the ability of the scabies PCR and microscopy to identify such cases. It would be better to discuss in terms of PCR having an ability to identify cases that could not be confirmed by microscopy.

2) Line 274-275: Did you statistically test whether there was a difference in the sensitivity for PCR and for microscopy? From what I can see, you tested whether there was a difference for each of these methods across the different IACS categories, but didn’t test the methods against each other. Please change the wording to ensure it is clear whether or not a formal statistical test was undertaken.

**Editorial and Data Presentation Modifications?**

Reviewer #1: Writing of the references should be checked

Reviewer #2: Minor corrections and suggested re-phrasing:

1) Line 28: ‘Sarcoptes’, not ‘Scarcoptes’

2) Line 30: ‘proposed by the International…’

3) Line 31: ‘tests’ not ‘test’

4) Line 32: as no specific PCR test has yet been mentioned, ‘this in-house…’ does not make sense

5) Line 62: ‘scabiei’, not ‘scabies’

6) Line 74: delete ‘is’

7) Line 76: ‘result in various skin eruptions…’

8) Line 87: ‘tests’, not ‘test’

9) Line 145: ‘were performed’

10) Line 158: ‘scraping’, not ‘scrapping’

11) Line 197-198 seems to simply repeat the preceding sentence.

12) Line 203: 10/30 = 33%, not 30%

13) Line 214: for confirmed, clinical and suspected, it should be 22/30 (73% is correct), not 24/30

14) Line 224: ‘on the other contrary’ sounds strange – how about ‘on the other hand’?

15) Line 261: ‘The authors’, not ‘they’

**Summary and General Comments**

Reviewer #1: --

Reviewer #2: Thank you for the opportunity to review this manuscript in which the authors aim to compare the sensitivity and specificity of a new PCR diagnosis tool for scabies to another commonly used method, microscopic confirmation. To do this, the authors use as the ‘gold standard’ the consensus criteria recently established by IACS, which appears to be a novel method, as previous studies treat either PCR or microscopy as the ‘gold standard’ and compare the other method to it. The authors claims that PCR testing may be useful in the diagnosis of scabies, due to its increased sensitivity over microscopy, are justified by the research study. Any improvement in the ability to diagnose scabies will be an important and relevant advance to both researchers and practitioners. There is some work to be done to ensure accuracy of numbers used throughout the paper, however, the conclusions would not change. While the paper is generally well written, I have indicated some typographical errors I noticed during review.

PLOS authors have the option to publish the peer review history of their article (what does this mean?). If published, this will include your full peer review and any attached files.

Reviewer #1: Yes: Aysegul TAYLAN OZKAN

Reviewer #2: No

---

## [Decision Letter · Decision Letter 1]

17 Mar 2020

Dear Dr. Kim,

We are pleased to inform you that your manuscript 'Diagnostic value of the molecular detection of Sarcoptes scabiei from a skin scraping in patients with suspected scabies' has been provisionally accepted for publication in PLOS Neglected Tropical Diseases.

Best regards,

Kosta Y. Mumcuoglu, PhD

Associate Editor

Paul J Brindley, PhD

Editor-in-Chief

None

Reviewer's Responses to Questions

**Key Review Criteria Required for Acceptance?**

**Methods**

-Are the objectives of the study clearly articulated with a clear testable hypothesis stated?

-Is the study design appropriate to address the stated objectives?

-Is the population clearly described and appropriate for the hypothesis being tested?

-Is the sample size sufficient to ensure adequate power to address the hypothesis being tested?

-Were correct statistical analysis used to support conclusions?

-Are there concerns about ethical or regulatory requirements being met?

Reviewer #1: The author made all necessary changes

Reviewer #2: (No Response)

**Results**

-Does the analysis presented match the analysis plan?

-Are the results clearly and completely presented?

-Are the figures (Tables, Images) of sufficient quality for clarity?

Reviewer #1: The author made all necessary changes

Reviewer #2: (No Response)

**Conclusions**

-Are the conclusions supported by the data presented?

-Are the limitations of analysis clearly described?

-Do the authors discuss how these data can be helpful to advance our understanding of the topic under study?

-Is public health relevance addressed?

Reviewer #1: The author made all necessary changes

Reviewer #2: (No Response)

**Editorial and Data Presentation Modifications?**

Reviewer #1: The author made all necessary changes

Reviewer #2: There is a small number of typographical errors to be corrected (line numbers are from the marked-up word version of the manuscript):

Line 8: 'Dermatology' not 'Dermartology'

Lines 51-52: either 'tests...have' or 'test...has'

Line 76: 'result' (to match with 'Infections')

Line 242: ...'according to gender...'

Line 285-287: sentence is incomplete

References 4 & 10 have inconsistent capitalisation of the title

**Summary and General Comments**

Reviewer #1: The author made all necessary changes

Reviewer #2: Thank you for the opportunity to re-review this manuscript. I am satisfied that all of my comments have been addressed, and I thank the authors for the clear manner in which they have presented the changes, which saved time in re-review.

PLOS authors have the option to publish the peer review history of their article (what does this mean?). If published, this will include your full peer review and any attached files.

Reviewer #1: Yes: Aysegul TAYLAN-OZKAN

Reviewer #2: No

---

## [Editor Report · Acceptance letter]

1 Apr 2020

Dear Dr. Kim,

We are delighted to inform you that your manuscript, "Diagnostic value of the molecular detection of Sarcoptes scabiei from a skin scraping in patients with suspected scabies," has been formally accepted for publication in PLOS Neglected Tropical Diseases.

Best regards,

Serap Aksoy

Editor-in-Chief

Shaden Kamhawi

Editor-in-Chief
